# Position: Biology is the Challenge Physics-Informed ML Needs to Evolve

**Julien Martinelli**
ELLIS Institute Finland
Department of Computer Science, Aalto University
julien.martinelli@aalto.fi

## Abstract

Physics-Informed Machine Learning (PIML) has successfully integrated mechanistic understanding into machine learning, particularly in domains governed by well-known physical laws. This success has motivated efforts to apply PIML to biology, a field rich in dynamical systems but shaped by different constraints. Biological modeling, however, presents unique challenges: multi-faceted and uncertain prior knowledge, heterogeneous and noisy data, partial observability, and complex, high-dimensional networks. **In this position paper, we argue that these challenges should not be seen as obstacles to PIML, but as catalysts for its evolution. We propose Biology-Informed Machine Learning (BIML): a principled extension of PIML that retains its structural grounding while adapting to the practical realities of biology.** Rather than replacing PIML, BIML retools its methods to operate under softer, probabilistic forms of prior knowledge. We outline four foundational pillars as a roadmap for this transition: uncertainty quantification, contextualization, constrained latent structure inference, and scalability. Foundation Models and Large Language Models will be key enablers, bridging human expertise with computational modeling. We conclude with concrete recommendations to build the BIML ecosystem and channel PIML-inspired innovation toward challenges of high scientific and societal relevance.

## 1 Introduction

Physics-Informed Machine Learning (PIML [36, 49, 90]) has emerged as a powerful modeling paradigm that blends mechanistic insight with data-driven flexibility. By embedding known physical laws, typically expressed as differential equations, into machine learning systems, PIML has achieved impressive results in domains like fluid dynamics [32], climate science [91], and materials modeling [104]. These successes build on favorable conditions: systems governed by well-understood equations, often paired with relatively structured and time-resolved data, where the modeling challenge is accelerating simulations rather than scientific discovery.

This success has spurred interest in applying similar methods to other domains with time-resolved dynamics. Biology, at first glance, offers a parallel opportunity: many biological systems evolve over time in response to environmental or internal cues [50]. A core aim of systems biology is to uncover how genetic [5], metabolic [106], or ecological mechanisms drive behavior across scales. Ordinary Differential Equations (ODEs) have long been used to capture these dynamics, modeling how concentrations of molecules like proteins or drugs evolve according to reaction kinetics [65].

While this convergence holds the promise of transferring PIML to biological dynamical systems, the analogy breaks down under closer inspection. The majority of problems tackled by PIML to date, primarily in physics and engineering, feature systems that are fully observable, governed by known equations, and measured with relatively low noise. These characteristics have shaped the field's

development. In contrast, biological knowledge is often qualitative, fragmented, or context-dependent, and rarely manifests as governing equations. Measurements are sparse, noisy, and heterogeneous across individuals, species, and perturbations, and many relevant components remain unobserved or unmeasurable. Rather than low-dimensional models, biological systems typically involve large, nonlinear feedback networks with dozens or hundreds of interacting variables [2, 5, 26, 110].

Despite its conceptual appeal, PIML has seen limited uptake in biology. Its use has largely focused on accelerating simulations of known Partial Differential Equations from physics, whereas biology demands tools for uncovering and modeling complex dynamical systems, often expressed as ODEs. Crucially, a salient point within the Machine Learning (ML) community is the absence of benchmarks tailored to biological dynamical systems modeling. While such a gap is understandable, it currently prevents systematic assessment of PIML's effectiveness or its comparison to emerging alternatives.

**This position paper argues that PIML must evolve to meet the unique challenges of biological modeling, and that this evolution represents not a limitation, but a major opportunity, giving rise to Biology-Informed Machine Learning (BIML).** This transition is a chance to reimagine how the integration of various forms of knowledge, data-driven approaches, and recent advances in Foundation Models (FMs), including Large Language Models (LLMs), can come together to tackle the distinct challenges of biology. We outline the conceptual and methodological shifts needed to support this transition and offer concrete recommendations to build the community, infrastructure, and benchmarks required to make BIML a reality. Our goal is to initiate a field-wide conversation on how PIML can rise to meet the scientific and epistemic demands of biology.

## 2  Why Physics-Informed ML Falls Short in Biology

**What is Physics-Informed Machine Learning?**

At its core, Physics-Informed Machine Learning integrates prior scientific knowledge, typically in the form of differential equations, directly into ML models. This is most commonly achieved by enforcing physical constraints during training, such as penalizing deviations from known dynamics through additional loss terms. It results in a class of models that are not only data-efficient but also grounded in mechanistic theory, offering improved generalization and interoperability.

$$\mathcal{L}_{\text{total}} = \underbrace{\frac{1}{N}\sum_{i=1}^{N}\|f_\theta(x_i) - y_i\|^2}_{\text{data mismatch for ML model } f_\theta} + \lambda_{\text{phys}} \cdot \underbrace{\frac{1}{M}\sum_{j=1}^{M}\|\mathcal{F}[f_\theta](x_j^{\text{phys}})\|^2}_{\text{physics mismatch w.r.t governing equation } \mathcal{F}}$$

While PIML is often used to accelerate simulations of known systems, we adopt a broader view, as a flexible modeling paradigm that blends mechanistic insight with data-driven inference in the service of scientific discovery [34, 60, 86].

In what follows, we identify four recurring structural mismatches between PIML's modeling assumptions and the realities of biological data, shown in Figure 1. We also discuss the critical absence of principled benchmarks tailored to the complexities of biological systems.

### 2.1  Challenges

**Challenge 1: Multi-faceted and uncertain prior knowledge.**

A defining feature of PIML is the ability to encode mechanistic knowledge into model structure, typically in the form of differential equations from first principles [49]. In physics and engineering, this is often feasible: governing laws are well-established and widely accepted, like the Navier–Stokes equations for fluid dynamics. Under such conditions, strong theoretical guarantees, such as consistency and faster convergence, can be proven [22, 21]. But even slight misspecification in the equations or their parameters breaks these guarantees. Some methods, like SINDy [9], relax that assumption by selecting dynamics from a predefined library of candidate kinetics. While this introduces structural flexibility, such methods rarely support expressing or managing uncertainty about which kinetic forms are biologically plausible in a given context [45].

Translating biological knowledge into differential equation models is inherently difficult: it requires converting qualitative, often informal information into precise mathematical structure. Unlike

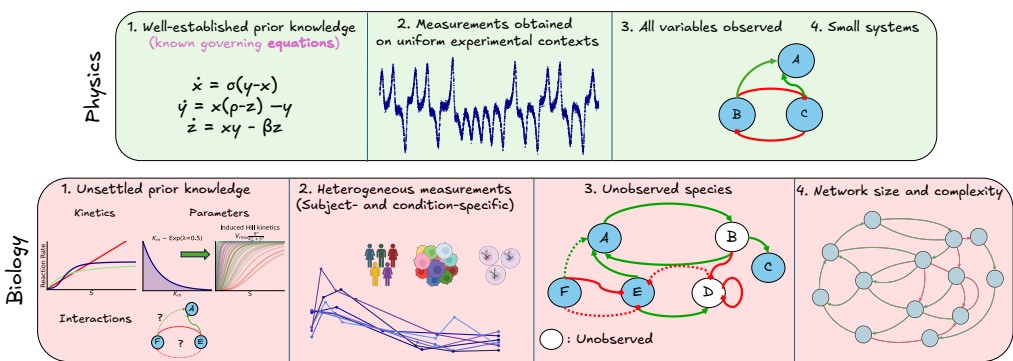

Figure 1: **Biology-specific challenges in dynamical systems discovery.** While the field has mostly focused on problems arising from physics (top panel), the resulting methods are not geared towards the unique challenges inherent to biological data (lower panel).

physics, biological understanding is distributed across diverse sources—curated pathway databases, empirical studies, and textual annotations—with varying levels of completeness and reliability. Even basic structural information, such as which molecular species interact, is often uncertain or only partially known. In some cases, high-level properties like sparsity or network topology can be assumed [6, 84]; in others, specific interactions are suggested by resources like KEGG [48], Reactome [77], STRING [109], or BIOGRID [83], though these vary in coverage and consistency.

More broadly, current PIML approaches lack mechanisms to integrate uncertain or semi-structured knowledge. Without rethinking how prior knowledge is represented, weighted, and integrated, PIML methods will remain limited to toy biological settings where assumptions can be artificially enforced.

**Challenge 2: Data Heterogeneity.** Biological data rarely conforms to the structure seen in the physical domains that shaped most PIML frameworks: clean, time-resolved trajectories of a single, well-defined system. Instead, biological datasets typically consist of measurements collected across multiple individuals, conditions, or perturbations, each reflecting variation in genetic background, environment, disease state, or experimental intervention [12, 108, 126]. This heterogeneity is not noise to be averaged out, but a fundamental feature of the system that must be modeled explicitly.

A few recent PIML methods address heterogeneity through context-aware modeling [82, 85], but typically assume fixed functional forms across environments. This setting breaks down in biological systems: heterogeneity often runs deeper, involving not just parameter shifts but structural differences. The key question is often not just what the system's dynamics are, but how they vary across contexts, and which structural elements remain invariant. For example, motifs like negative feedback or feedforward loops may be conserved across cell types or species [2], even as their parameters or downstream effects differ. Biology demands models that capture both shared and context-specific components of a dynamical system, and link their variation to biological function.

**Challenge 3: Unobserved species.** PIML methods often assume full or near-complete observability to constrain governing equations, even though there is extensive work on state estimation for partially observed physical and networked systems (e.g., climate, cosmology, materials); such observability is common in physics but rare in biology, where many key species are never observed directly and available readouts are indirect. As a result, model identification becomes ill-posed, requiring frameworks that treat partial observability as the norm and reason explicitly about hidden components. Many key variables in biological systems are unmeasured or unmeasurable: intracellular concentrations, regulatory factors, signaling intermediates, and compartmentalized dynamics often lie beyond experimental reach due to technical, ethical, or cost constraints. This challenge extends beyond molecular species, contextual factors like microenvironments or disease progression stages may reflect latent processes that nonetheless shape system behavior [55, 111].

**Challenge 4: Network Size and Complexity.** PIML methods were not built for high-dimensional systems; in physics, models are typically low-dimensional with known structure, enabling strong priors to guide learning. In biology, by contrast, both the system structure and parameters are typically unknown, and the number of potential interactions grows combinatorially, overwhelming methods like SINDy or symbolic regression [9, 98]. Biological networks often involve dozens or hundreds of

interacting species ([54, Figure 6], [92, Figure 4]), connected by nonlinear, feedback-rich interactions. This complexity is not incidental: it reflects the robustness of biological processes [118, 122], achieved in part through redundancy [42]. These systems are shaped to remain functional under environmental noise, molecular fluctuations, and structural perturbations, necessitating dense interconnections and dynamic compensation mechanisms [51]. Addressing this requires scalable inference and principled constraints on the model space to preserve interpretability and fidelity.

## 2.2 No Established Biological Benchmarks for PIML

The four challenges outlined above are not theoretical edge cases. They are defining characteristics of real biological systems. Yet critically, these features are almost entirely absent from the benchmark datasets that underpin most PIML development and evaluation.

Canonical benchmarks for dynamical systems identification, such as ODEBench [18] and SR-Bench [43], are primarily rooted in physics and engineering. They focus on low-dimensional systems with synthetic data that is fully observed, densely sampled, and minimally noisy. Regarding size, for instance, ODEBench includes 63 systems, but only 12 have dimensionality $D \in \{3, 4\}$, and just 2 of those are biologically motivated; the rest are simpler ($D < 3$). The more recent LLM-SRBench introduces 28 biology-related examples, but all are one-dimensional and synthetically constructed by combining canonical terms from the literature with artificial variations to encourage novelty [103].

While useful for method development, existing benchmarks reflect idealized settings and diverge from the ambiguity and partial observability of real biological data. Some PIML approaches have been applied to moderately complex systems like glycolytic oscillators [68, 82], but typically under conditions involving hundreds of synthetic trajectories with varied initialization, a regime rarely seen in practice. Learning dynamics under realistic, low-data conditions remains an open challenge [74].

As a result, we currently lack principled ways to evaluate how PIML methods perform under the complexities of biological systems. This limits progress and risks overstating the generality of approaches validated only on idealized tasks. To move forward, the field must rethink its evaluation paradigm by developing benchmarks that explicitly stress-test methods under biological conditions. Without it, PIML risks stalling at the edge of biological relevance.

## 3 The Biology-Informed ML Paradigm

The expression *biology-informed machine learning* has recently gained traction, especially in biomedical settings where prior knowledge such as pathways or interaction networks is integrated into ML models [23, 128]. Reviews document this trend in oncology [121] and in systems biology [87]; a survey of digital-twin learning from biological time series argues for hybrid, modular approaches that couple mechanistic structure with uncertainty quantification and modern generative tools [81]. Most work still targets static omics; applications to biological dynamics remain limited, with notable exceptions that apply neural ODEs in pharmacology [10, 88, 112, 130]. A bibliometric analysis also shows a marked rise of PIML in the biomedical literature (Figure 3, Appendix A). This position paper advances a perspective on BIML centered on dynamical systems and its relationship to PIML.

> **Definition (Biology-Informed Machine Learning)**
>
> Biology-Informed Machine Learning is a novel modeling paradigm that extends the core principle of Physics-Informed Machine Learning by integrating multi-source, informally encoded, and uncertain biological knowledge into data-driven modeling of dynamical systems.

BIML embraces epistemic and practical constraints, integrating diverse biological information, including partially known interactions, context-dependent structures, and learned representations of heterogeneity, into ML workflows for scientific discovery in biological dynamical systems. This goes beyond prior uses of the term, often limited to feature structuring in static omics datasets. Section 2.1 outlined the core challenges posed by biological data; we now turn to how BIML can address them.

### 3.1 The four pillars of BIML

To address the 4 challenges from Section 2.1, BIML emphasizes 4 methodological pillars: uncertainty quantification, contextualization, constrained latent structure inference, and scalable modeling. FMs

and LLMs are expected to support all pillars by providing biologically grounded priors, proposing plausible latent mechanisms, and efficiently integrating domain knowledge from literature, databases, and expert inputs. While full methodological proposals are beyond this position paper's scope, we elaborate on each pillar to outline impactful research directions and guide future work.

**Pillar 1: Uncertainty Quantification.** Due to the ambiguity of biological knowledge, where priors vary in confidence, origin, and context, BIML places uncertainty quantification at its core. For example, a regulatory interaction between two molecular species may appear in one database but not others, reflecting both a lack of consensus and variation in interaction type, such as transcriptional regulation, protein-protein binding, or post-transcriptional inhibition [69, Figure 1.1]. These differences in granularity often result in conflicting or incomplete views across resources. Probabilistic frameworks, particularly Bayesian approaches, offer a natural mechanism for handling such discrepancies by treating model structure and parameters as distributions conditioned on data and prior beliefs [29, 96, 101]. This is key in high-stakes settings such as therapeutic design, drug repurposing, or personalized medicine, where acting on incorrect assumptions can have costly consequences.

Quantifying uncertainty allows researchers to assess how much predictions depend on assumed interactions, kinetics, or parameters, and to prioritize data collection or decision-making accordingly [89]. Gaussian-process–based constrained modeling in biology has a long lineage, including ODE-constrained and gradient-matching approaches that already confronted noise and partial observability in small systems [11, 20, 62]. Recent physics-informed GP variants [61] and probabilistic Neural ODEs [97] continue this trajectory, yet a BIML approach must go further, embedding uncertainty throughout the entire modeling stack. This includes for instance parametric uncertainty, tied to unknown or poorly constrained mechanistic parameters such as reaction and degradation rates; structural uncertainty, arising from incomplete or conflicting prior knowledge about dynamic interactions; epistemic uncertainty, reflecting gaps in available data; and aleatoric uncertainty, due to intrinsic biological variability and measurement noise. A recent example from pharmacokinetics employs manifold-constrained Gaussian processes for mixed-effects ODE models, yielding subject-level uncertainty for trajectories and quantities of interest such as peak and trough concentrations [129].

Among BIML's foundational principles, this first pillar may be the most critical: without a principled account of uncertainty, we risk drawing strong conclusions from weak or inconsistent evidence, a pervasive danger in complex biological systems.

**Pillar 2: Contextualization.** Biological data are inherently heterogeneous, varying across individuals, tissues, perturbations, and conditions. BIML moves beyond one-size-fits-all modeling, introducing frameworks capable of disentangling shared mechanisms from context-specific variations. This pillar is closely tied to uncertainty: when data are pooled across diverse contexts, uncertainty quantification becomes essential to distinguish robust, generalizable patterns from spurious or condition-specific effects. Mixed-effects models offer a principled approach for encoding population-level structure while allowing for individual-level deviations [58]. They have been successfully amortized [3] and extended to the functional setting [59, 63], though not yet in the context of dynamical systems modeling. Similarly, multi-task learning and hierarchical modeling provide mechanisms for knowledge sharing across conditions while preserving flexibility. Recent advances in meta-learning, such as context-conditioned neural modules or embeddings, can further enable rapid adaptation to new experimental settings [41, 80].

**Pillar 3: Constrained Latent Structure Inference.** Biological systems are intrinsically partially observed. Latent processes in biology may stem from unmeasured molecular species [73, 39], transient intermediates, compartment-specific signals, or contextual factors such as microenvironments and disease stages. BIML will therefore adopt strategies that embrace partial observability as the norm. Although existing methods like structured latent variable models, hybrid neural-mechanistic frameworks [33, 88], and inference techniques to recover hidden dynamics from incomplete trajectories [95, 105] offer important starting points, BIML demands a fundamentally different treatment of latent structure. What distinguishes BIML is that latent variables are not introduced for expressive convenience, but represent hypothetical yet interpretable biological components, entities whose existence or influence may be uncertain but biologically motivated. As such, BIML must explicitly quantify uncertainty over their presence and role, extending the principles of Pillar 1 into the latent space. In parallel, these components should be constrained by prior knowledge, such as interaction networks, spatial organization, or pathway ontologies. This grounding is essential to avoid introducing latent variables as abstract statistical constructs unanchored in biological interpretation.

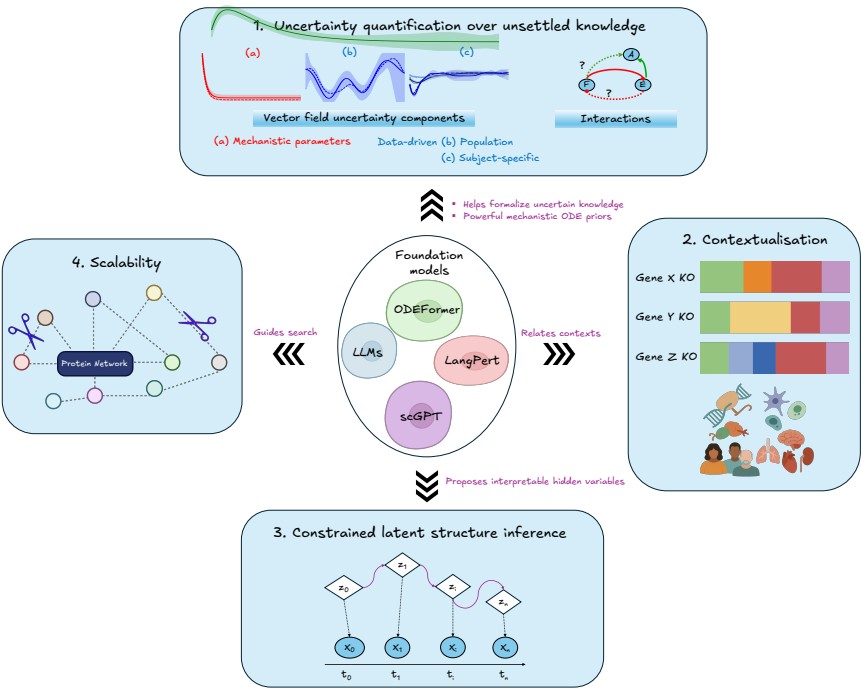

Figure 2: **The four pillars of Biology-Informed Machine Learning and the integrative role of Foundation Models.** FMs and LLMs support each BIML pillar by embedding biological knowledge, guiding inference, and enabling scalable, uncertainty-aware modeling across heterogeneous and partially observed systems.

**Pillar 4: Scalability.** To handle the dimensionality of real-world biological systems, BIML must achieve computational efficiency without compromising model fidelity or uncertainty quantification, both of which are critical in high-stakes downstream tasks like healthcare. This calls for the development of tailored inference strategies, such as last-layer Bayesian neural networks [37] variational Gaussian score matching [79], random fourier features for Physics-Informed Gaussian Processes [38], and sampling-based methods like subset-weighted tempered Gibbs sampling [44], which offer tractability in high-dimensional regimes. Scalability also hinges on model architecture. Structured priors, sparse or low-rank representations, and modular designs help constrain the hypothesis space, enhancing both interpretability and data efficiency as complexity grows. As an example, adopting a chemical reaction network (CRN) perspective on biological dynamics enables a structured decomposition of the system into interpretable reaction motifs. This framing allows the ODE inference problem to be reformulated as CRN structure discovery [45], which can leverage scalable strategies such as sequential or modular inference over candidate reactions [56, 74, 75].

## 3.2 Foundation Models and Large Language Models as an Integrative Layer in BIML

FMs and LLMs offer a compelling infrastructure to expedite BIML. By encoding semi-structured biomedical knowledge and leveraging massive datasets, they have already enabled the generation of biologically grounded priors [13, 16, 35]. While such models do not yet explicitly incorporate temporal dynamics, FMs for ODEs [18, 99] have recently emerged, offering new opportunities for biological dynamical systems. For instance, ODEFormer [18] predicts symbolic ODE terms from data, enabling interpretable dynamics modeling with minimal supervision. Overall, these FMs inform mechanistic discovery and data-efficient modeling workflows. They support all pillars (Figure 2) by:

**(Pillar 1)** Assisting in formalizing and weighting uncertain prior knowledge, as demonstrated by tools like LLM-Lasso [125] and Al-Khwarizmi [80].

**(Pillar 2)** Helping interpret and disambiguate heterogeneous contexts [35, 72, 124], enabling extrapolation to unseen conditions such as molecular perturbations [66]. As example, LangPert [72] uses LLMs to extrapolate perturbation responses by inferring latent context from partial information.

**(Pillar 3)** Proposing biologically plausible latent components and assess their epistemic plausibility.

**(Pillar 4)** Informing scalable model construction by suggesting modular structure, pruning implausible interactions, and constraining search over reaction-level dynamics [102].

These examples indicate that foundation models can serve as *infrastructure* in BIML workflows: scaffolds for prior elicitation and knowledge grounding, hypothesis generation, and linking unstructured biological context to mechanistic models. They are complements to PIML/BIML, not replacements for mechanistic inference or statistical calibration. Predictive gains are not automatic. On several tasks simple baselines still outperform LLMs, including perturbation response prediction [117, 119], even if careful metric design can affect these comparisons [76]. High-dimensional numerical data remain difficult due to tokenization and representation constraints [46]. Finally, ODE-focused FMs like ODEFormer will need to be carefully tailored to biological dynamical systems by incorporating domain-specific datasets or enforcing structural constraints such as conservation laws.

Beyond static knowledge integration, LLMs can support interactive reasoning under uncertainty by suggesting refinements and synthesizing mechanistic hypotheses as new data arrive. We advocate virtual laboratory infrastructures: modular, AI-assisted environments for simulation, experimentation, and model refinement under biological constraints [52, 100]. In this setting, LLMs act as an interactive modeling assistant that surfaces relevant literature, queries mechanistic databases, validates constraints, and guides experiment design; for example, Lab-in-the-Loop demonstrates iterative therapeutic antibody design with model-suggested experiments [27]. We use AI as the bridge between physics and biology: a shared interface where physicists encode mechanisms and invariants, biologists supply context and measurements, and the modeling layer reconciles both into priors, tests them under uncertainty, and returns interpretable artifacts for the next iteration. Crucially, these tools engage practitioners directly, enabling adaptive workflows that incorporate their input and respond to evolving hypotheses, while treating experts as full collaborators with their own strategic goals [14].

Rather than treating biology as a pathological case for existing PIML tools, we argue it should serve as a catalyst for ML innovation. Its complexity challenges standard assumptions and demands new abstractions, representations, and workflows. In this vision, LLMs are not just accelerators; they expand modeling abstraction, enabling deeper engagement with the scientific process. Given their tendency to hallucinate in underspecified or ambiguous settings [40], which are commonplace in biology, ensuring their factual reliability is a critical area for future work. This can be mitigated by leveraging expert feedback and Retrieval Augmented Generation [30].

### 3.3 Illustrative example: Gene Regulatory Network inference as a BIML use case

Inferring a gene regulatory network involves recovering the set of interactions by which genes regulate each other's expression over time. This task is central to systems biology and typically relies on temporal gene expression data, which may come from bulk RNA-seq experiments or pseudo-time single-cell RNA-seq processed using trajectory inference tools [4]. Using this limited, noisy temporal data across multiple cell lines and perturbations, BIML would approach this problem as follows:

---

**Gene Regulatory Network inference through the lens of BIML**

- **(P1)** Construct uncertain priors over network structure and kinetics using curated databases, expert input, and LLM-assisted literature synthesis. ODEFormer [18] suggests symbolic dynamics, LLM-Lasso [125] assigns informative regularization weights; candidate mechanisms surfaced by LLMs are curated by human experts. Bayesian model averaging represents structural uncertainty.

- **(P2)** Contextual variation across cell lines or perturbations is embedded using representations from LLMs and FMs trained on biomedical corpora. Dedicated pretrained single-cell models like scPRINT provide cell-state embeddings learned with GRN-oriented objectives and can serve as context features [47]. These embeddings can inform Gaussian Process priors that capture condition-specific deviations while sharing structure across conditions [71].

- **(P3)** Many regulators are unmeasured. We introduce latent variables in the ODE to represent hidden transcription factors and use LLMs to suggest plausible candidates from the literature. Human modelers validate or reject these suggestions as part of the inference workflow.

- **(P4)** The model space is large. Scalable inference methods are combined with modular priors and LLM-assisted pruning of implausible interactions to reduce search complexity.

---

Evaluation can consider held-out conditions and perturbations; report edge-level precision/recall against curated resources and perturbation ground truths; and probe counterfactual validity via *in silico* knockouts and dose changes, using biological robustness as a qualitative hallmark [51]. Simple baselines and targeted ablations help isolate the value of priors, context embeddings, and latent structure. These guidelines are a starting point rather than a prescription. Appendix B details a second use case, on generalization to unseen interventions.

# 4 Recommendations for a Successful Transition to BIML

Beyond methodological developments outlined above, we propose actionable steps to expedite BIML.

## 4.1 Rethinking Benchmarks for Biological Dynamical Systems

As discussed in Section 2.2, existing benchmarks for dynamical system modeling are largely physics-inspired and fail to reflect key features of biology. Without such realism, it becomes difficult to assess whether PIML methods are appropriate for biological applications, or how they trade off robustness, interpretability, and generalization. This skews development toward methods optimized for idealized settings, creating a false sense of generality. Addressing this gap is essential to retooling PIML for biology and establishing BIML as a credible evolution of the paradigm.

We call for a new class of benchmarks, aligned with the realities of biological modeling. These should act as epistemic stress-tests: tools for uncovering failure modes, not just optimizing performance. They should probe system identification under uncertain priors (e.g., gene regulatory networks [70] or reaction kinetics [45, 74]), and predictive modeling under data heterogeneity, latent contextual variation, or unseen perturbations [72]. Curated synthetic datasets, such as those from BioModels [67], can serve as controlled settings for evaluating recovery and estimation, while real-world datasets can stress-test robustness, generalization, and uncertainty quantification, even in the absence of full ground truth. Without such realism, benchmarks risk incentivizing reward hacking—a term borrowed to Reinforcement Learning [28, 116]—where models overfit to narrow metrics or synthetic artifacts while failing to address the true scientific complexity of biological systems. As a result, evaluation metrics should reflect the multifaceted objectives of biological modeling, including robustness, generalizability, and alignment with biological knowledge.

Despite their importance, integrating such benchmarks into the ML development cycle remains a cultural and institutional challenge. There is a persistent reluctance to engage with the full complexity of real biological systems. These are precisely the settings where innovation is most needed, yet they are often avoided because they resist clean comparisons or performance gains. This hesitancy reflects an implicit bias: a preference for method-centric progress over domain-grounded relevance. But if machine learning is to grow into a biologically relevant science, benchmarks must evolve from performance validators into epistemic instruments. They must challenge assumptions, reveal blind spots, and serve as tools for falsification and learning, not just leaderboard updates.

## 4.2 From PIML to BIML: Embracing an Application-Driven View

The call for biologically grounded benchmarks reflects a deeper issue: machine learning progress, including PIML, has often prioritized methodological elegance over domain relevance. An application-driven view of ML [93] advocates building and judging models based on the structure of real-world problems. It urges us to ask: are we solving domain-relevant problems and evaluating models under realistic constraints? Just as biodiversity-focused datasets [113] and robustness benchmarks [53] have catalyzed progress on distribution shift and generalization, biology-centric benchmarks can similarly uncover blind spots and promote innovation tailored to biological complexity.

This reframing is especially urgent in high-stakes settings such as health and policy, where misaligned abstractions and opaque modeling choices can amplify biases, mislead stakeholders, and erode trust [7, 31]. Transitioning to BIML is thus not about abandoning PIML, as it remains invaluable where its assumptions hold. It is about shifting perspective: from designing methods to fit benchmarks, to designing benchmarks and methods aligning with the scientific questions we seek to answer.

### 4.3 Establishing BIML: Community, Collaboration, and Culture

To make BIML a lasting research direction, we must foster both interdisciplinary collaboration and targeted community-building. This includes organizing workshops, special sessions, and challenges at major ML conferences that foreground the unique demands of biological systems. Incentive structures must evolve accordingly: reviewing norms should explicitly value contributions that introduce biologically realistic datasets or expose method limitations under realistic conditions. Tracks such as NeurIPS Datasets and Benchmarks are natural venues, but stronger guidance is needed to ensure that biologically grounded work is not sidelined in favor of algorithmic novelty.

Cross-disciplinary collaboration is equally critical. Applying ML to biological systems requires close engagement with domain experts: biologists, pharmacologists, clinicians. This ensures that models target meaningful dynamics, generate actionable predictions, and quantify the right forms of uncertainty. BIML is not just a shift in methodology, but in workflow: a more interactive and context-sensitive approach to scientific modeling. Importantly, this shift may also influence experimental practice. As BIML methods emphasize temporal dynamics and causal reasoning, they can motivate practitioners to move beyond static snapshots and generate richer, time-resolved and perturbation-aware datasets. In this way, modeling and experimentation can become mutually reinforcing.

## 5 Alternative Views

We now address three key counterarguments to our position: whether incremental improvements to PIML are sufficient to meet the demands of BIML, whether PIML is the right foundation for modeling biological systems, and whether biology is the most worthwhile focus for PIML research among competing application domains.

**Incremental fixes to PIML are not enough.** A common view is that the challenges of biological modeling—uncertain prior knowledge, latent structure, heterogeneity, and scale—can be addressed through incremental extensions of PIML: better uncertainty quantification, more flexible priors, and scalable inference. Such improvements are valuable but not sufficient. The settings where PIML has thrived—well-specified dynamics, full observability, and clean data—are not occasionally absent in biology; they are systematically misaligned. Retrofitting PIML without addressing this mismatch risks brittle, narrowly applicable solutions. Progress will require not just refinement, but reframing.

**Why PIML remains a valuable foundation.** Given these misalignments, one might ask whether PIML is even the right starting point for modeling biological dynamical systems. Should we not build a new paradigm entirely from scratch? While this is a valid concern, designing new frameworks from the ground up, both conceptually and computationally, is slow and rarely adopted in practice. PIML already provides a compelling starting point: a framework for integrating mechanistic structure into learning, grounded in interpretability and inductive bias. BIML builds on this foundation, adapting it to the realities of biology, where prior knowledge is uncertain, observability is limited, and context matters. Rather than discarding PIML, BIML retools its methods to work with softer, probabilistic, and multi-source forms of biological information. The goal is not to replace PIML, but to evolve it toward a modeling logic attuned to biological complexity.

**Why biology should be the next frontier for PIML.** A final objection is strategic: why prioritize biology for PIML, rather than domains like climate science, where validation is easier and physical constraints are better understood? This overlooks two points. First, biology also embodies the key PIML-relevant principles that are structure, sparsity and modularity, but in latent, context-specific, and multi-scale forms. Uncovering this structure is not a departure from PIML, but its natural evolution Second, biology is not just another domain: it offers unmatched potential for impact in health and the life sciences. Focusing on it advances ML, where better models can drive real scientific and societal progress, broadening PIML's reach and deepening its foundations. To back up this claim, we surveyed PIML beyond systems biology. Appendix C synthesizes this broader view, covering other biological subfields and domains beyond. Across these settings, extensions of PIML are domain tailored rather than uniform. BIML can learn from these patterns, and progress in BIML should in turn inform those fields. These examples point to a shift toward domain-informed hybrids that prioritize realism and calibration while preserving mechanistic backbones.

# 6  Discussion and Conclusion

This position paper argues that while PIML provides a strong foundation for integrating mechanistic structure into learning, its traditional scope, which assumes clean equations, full observability, and structured data, rarely applies in biology. Biological systems are shaped by fragmented knowledge, operate across multiple scales, and are only partially observed. These are not exceptions but defining features. To address this, we introduced Biology-Informed Machine Learning, an evolution of PIML that embraces the complexities of biological modeling. BIML rests on four pillars: uncertainty quantification, contextualization, constrained latent structure inference, and scalability. Each pillar is rooted in persistent mismatches between PIML methods and biological data. FMs and LLMs help operationalize these pillars by structuring knowledge, proposing hypotheses, and enabling data-efficient model design. The goal is not to replace PIML but to adapt its strengths, namely structure, interpretability, and inductive bias, to the realities of biology. Biological systems combine exceptional complexity and ambiguity, and **rather than being an obstacle, this makes biology the challenge PIML needs to evolve**. By engaging with these intricacies, BIML pushes PIML toward greater robustness, expressivity, and scientific maturity. Ultimately, methods alone are not enough. Progress requires rethinking our standards: benchmarks that expose failure modes, evaluation norms grounded in domain relevance, and incentives aligned with biological insight.

We hope this position catalyzes discussion on the design and scope of BIML. We invite the community to reflect on its feasibility, implementation, and scientific potential. Specifically:

- How can FMs and LLMs be integrated systematically to address biology-specific modeling gaps?
- What new abstractions are needed to enable scalable, interpretable, and uncertainty-aware inference in high-dimensional biological systems?
- How can BIML methods support translational applications in health, and what infrastructure is needed to enable effective collaboration between ML and domain experts?

BIML is a call to real-world engagement. It challenges the ML community to meet biology on its own terms, not to dilute scientific rigor, but to deepen it. In doing so, ML can advance our understanding of living systems and evolve toward greater robustness, relevance, and responsibility.

> **Call to Action: Building the BIML Ecosystem**
>
> ✔ **For ML Researchers** Move beyond idealized physics-style benchmarks by contributing a DREAM-style task that reflects real biological constraints, thereby embracing biology's unique challenges as drivers of innovation. When publishing, release code, a concise data card, a short metric checklist including counterfactual tests, and simple baselines.
>
> ✔ **For Domain Scientists** Help shape the modeling agenda with ML colleagues by scoping an initial question and articulating constraints, plausible priors, and sanity checks grounded in biology. Share what measurements are available or missing, typical noise levels, and examples of nonsensical outcomes, and co-refine evaluation criteria so success tracks what matters in practice.
>
> ✔ **For the Broader ML Community** Support venues, benchmarks, workshops, and challenges that review practices that value realism alongside innovation. Encourage camera readies to include a "biological validity" checklist. Pilot an author opt-in for domain co-review where authors request a domain check and ACs invite a domain co-reviewer.

# 7  Funding

This work was supported by the Research Council of Finland (Flagship programme: Finnish Center for Artificial Intelligence FCAI and decision 341763), EU Horizon 2020 (European Network of AI Excellence Centres ELISE, grant agreement 951847), and ELLIS Finland.

# 8  Acknowledgements

The author is grateful to Ayush Bharti, Rafał Karczewski, and Déborah Boyenval for extensive comments during the preparation of this manuscript, and to the anonymous reviewers for their constructive feedback.

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

# Appendix - Position: Biology is the Challenge Physics-Informed ML Needs to Evolve

## A    Bibliometric trends in physics-informed machine learning for biomedicine

We quantified yearly publication counts for *physics*-adjacent terminology used in machine learning contexts from 2015 to 2024 using **OpenAlex** (Works) and **PubMed** (E-utilities), restricted to a health/biology slice: PubMed inherently indexes biomedical literature, and OpenAlex counts were filtered by Life/Health concepts (e.g., "Life sciences" and "Health sciences") via concept filters. We report a single series that pools close physics terms and intersects them with ML terms: physics terms were *physics-informed*, *physics-constrained*, *physics-inspired*, and the acronym *PINN*; ML terms were *machine learning*, *deep learning*, *neural network*, *Gaussian process*, and *Bayesian*. Figure 3 displays these results.

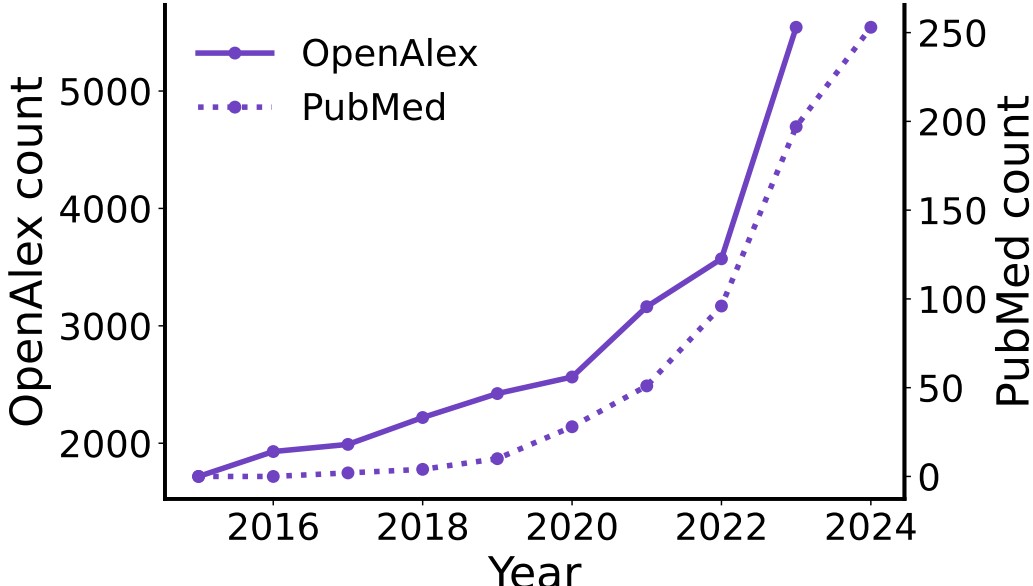

Figure 3: Growth of **physics-informed ML** terminology in biomedicine (2015–2024).

## B    Illustrative example of the BIML framework: generalization to unseen interventions

Predicting responses to unseen interventions such as new drugs or CRISPR knockouts requires extrapolation beyond observed conditions while remaining consistent with biological constraints. Recent benchmarking indicates that simple baselines can match or exceed complex deep models for post-perturbation RNA-seq prediction [15, 119, 76, 117], which suggests approaches that organize prior knowledge rather than relying on scale alone. Biology-Informed ML offers a path by pairing structured priors with contextual embeddings, constrained latent structure, and computational choices that keep inference tractable at biological scale.

- **(P1) Uncertainty quantification.** BIML could elicit uncertain priors on targets, pathways, and admissible kinetics from curated resources and literature distilled with LLMs, then propagate structural and parametric uncertainty to predictions and abstention. `CellBox` illustrates how kinetic structure can constrain extrapolation to unseen combinations [123], while `GEARS` shows that knowledge graphs can help simulate outcomes for genes and combinations not observed during training [94].

- **(P2) Contextualization.** A new intervention can be mapped into representation space with pretrained molecular and gene embeddings together with short textual synthesis. `LangPert` suggests that

LLM-derived gene context can improve leave-perturbation-out performance [72], and `scOTM` shows promise by fusing perturbation embeddings with single-cell data to transfer across cell types and assays [115]. Conditioning mechanistic or probabilistic cores on such embeddings could link novel drugs or knockouts to related mechanisms.

- **(P3) Constrained latent structure inference.** Complex interventions often trigger cascades with genuinely latent regulators and intermediates. BIML can treat these as hidden variables within ODE formulations and can constrain their role using admissible kinetics. LLMs may surface candidate hidden components and interaction hypotheses from the literature that experts curate for inclusion, after which latents and parameters are inferred jointly; kinetic constraints may improve identifiability and counterfactual consistency [123].

- **(P4) Scalability.** The hypothesis space grows with each new gene, compound, and context. BIML keeps computation tractable by combining modular priors with compositional representations of interventions, amortized inference that reuses shared components, and LLM-assisted pruning of implausible mechanisms; this reduces search and supports leave-perturbation-out and combinatorial scenarios at realistic scales.

We suggest leave-gene-out and leave-drug-out splits, calibration and counterfactual validity under dose or target swaps, and decision-relevant utility. Simple baselines should accompany BIML variants to quantify gains attributable to priors and structure. These ingredients are not a final solution. They sketch a pragmatic starting point for BIML implementations that can be adapted and expanded as tooling mature.

## C  BIML in context: adaptations across biology and beyond

Our main text developed Biology-Informed Machine Learning in molecular and systems biology as an extension of Physics-Informed Machine Learning that integrates uncertain and diffuse prior knowledge, explicit context, latent structure, and scalable modularity. Here we show that the same needs arise more broadly. We first survey biological subfields where process-based models are augmented with learned components under conservation and stoichiometric constraints. We then document analogous, domain-tailored hybrids beyond biology, including mobility, buildings, electric grids, and climate, where human and operational decisions or unresolved multi-scale physics force departures from vanilla PIML. Across these settings, practitioners begin with a physics-informed scaffold and add uncertainty handling, context encoding, constrained latents, and scalable composition so models remain useful under partial observability, heterogeneity, and distribution shift. This situates BIML within a cross-domain shift from purely physics-informed models to domain-informed hybrids that prioritize realism and calibration while preserving mechanistic backbones.

### C.1  BIML across diverse biological subfields

**Plant phenomics and controlled agriculture.**  Crop growth depends on transport physics and photosynthesis kinetics, yet outcomes vary strongly with genotype and environment. Hybrid pipelines retain mechanistic crop or transport modules and learn residual responses and context from sensor streams. In controlled aeroponics, a physics-based crop model coupled with ML improved prediction of biomass and leaf area and estimated resource use from IoT data; remaining errors on nitrate and water use highlighted data sparsity and physics gaps under unconventional conditions [24].

**Ecology and environmental biology.**  Process-based ecosystem models encode conservation laws and flux couplings but struggle with site-specific variability and sparse measurements. Recent work couples differentiable ecophysiological simulators with learnable components so that the simulator enforces mass and energy balance while neural terms learn uncertain parameters and residual processes. In photosynthesis and carbon–water coupling, these hybrids improve fit and transfer across plant types and conditions [1].

**Metabolic modeling and genome-scale constraints.**  Constraint-based genome-scale models predict phenotypes under media and genetic changes but often depend on labor-intensive uptake-flux assays. Neural–mechanistic hybrids embed stoichiometric and thermodynamic constraints while

learning residual mappings and context from data, improving growth-rate and knockout-phenotype prediction with far fewer labels than standard ML and without extra flux measurements [25].

**Biomechanics and physiology.** Classical inverse approaches for soft tissues are limited by multi-scale heterogeneity and the impracticality of *in vivo* internal stress measurements. Physics-informed learning reframes property identification as training neural fields that satisfy balance laws and constitutive relations while fitting observed deformations, enabling reconstruction of heterogeneous micromechanical properties in brain and heart valves with accuracy gains over standard pipelines [120].

**Convergent adaptations across subfields.** Relative to vanilla PIML that fits clean governing equations to observed fields, these efforts make recurring adjustments that mirror BIML's pillars. First, they admit diffuse, context-dependent priors and encode context explicitly: genotype and site factors in controlled agriculture [24] and site-specific drivers in ecosystem photosynthesis [1]. Second, they elevate unmeasured drivers to constrained latents, for example latent uptake and regulatory fluxes in genome-scale metabolism [25] and internal stresses or spatially varying material properties in soft tissues [120]. Third, they preserve mechanistic backbones while learning residuals only where theory is weak, which enables modular reuse and scalability across conditions [24, 1]. Together, these choices operationalize uncertainty quantification over mechanisms and parameters, contextualization, and constrained latent structure, while maintaining domain constraints such as stoichiometry, mass and energy balance, and constitutive laws [25, 1, 120].

## C.2 Physics-Informed ML Beyond Biology

**Urban mobility.** Classical traffic models encode conservation and fundamental-diagram relations at the macroscopic scale (Lighthill–Whitham–Richards, Aw–Rascle–Zhang) and car-following mechanics at the microscopic scale (Intelligent Driver Model). They underperform when human factors dominate: reaction-time variability, anticipation, rule-breaking, heterogeneous intent, probe-vehicle sampling bias, and GPS errors all induce misspecification. Recent adaptations embed these laws in a learnable architecture and fit residual behavioral terms and context from trajectories; in car-following, such hybrids improve accuracy under sparse data and out-of-distribution regimes relative to pure simulation or pure learning [78]. A recent survey synthesizes these designs, including neural components coupled to physics constraints, soft or hard enforcement of conservation, learned surrogates for fundamental diagrams, and uncertainty-aware training, and highlights treating driver intent as a constrained latent state [19].

**Urban energy systems.** Physics-based building models capture heat transfer and Heat, Ventilation and Air Conditioning dynamics but often miss site heterogeneity and human factors such as occupant schedules, window opening, and ad hoc set-point changes. Together with sensor noise, these effects create model misspecification in practice. Hybrid methods now marry gray-box physics with learning so that mechanistic submodels enforce energy balance and bounds while residual components fit building-specific behavior; for infiltration (air leakage), this improves accuracy and extrapolation to new operating conditions [127]. A recent review systematizes physics-informed inputs, losses, architectures, and simulation-to-measurement adaptation, reporting gains in small-data and partial-knowledge regimes where operational variability dominates [64].

**Electric power systems and grids.** Classical grid models encode network physics and control, yet real systems are non-stationary and only partially observed. Weather-driven renewables, market dispatch on 15–60 minute blocks, and consumer behavior cause parameters to drift and invalidate fixed-coefficient assumptions. Recent work couples a stochastic differential model of load–frequency dynamics with a neural mapping from techno-economic context (generation mix, ramps, prices) to time-varying parameters, yielding calibrated probabilistic forecasts and improved system identification on continental-scale data [57]. Complementary Bayesian physics-informed neural networks target inverter-dominated networks and quantify posterior uncertainty while maintaining physical consistency [107].

**Weather and climate.** Numerical models encode conservation and governing dynamics but struggle with uncertain sub-grid processes and multi-scale feedbacks, which leads to missed extremes and short-lead bias. Hybrid approaches embed physical evolution constraints within learned models and

treat unresolved processes as constrained latent components with explicit uncertainty. For extreme-precipitation nowcasting, a physics-conditioned deep generative system that enforces mass-conserving evolution outperforms a state-of-the-art numerical weather prediction baseline on stakeholder-relevant events [17]. A broader synthesis emphasizes that reconstruction, parameterization, and prediction improve when symmetries, balances, and conservation are preserved and uncertainty is quantified for assimilation and decisions [8]. Continuous-time neural models that implement value-preserving transport via physics-informed neural ODEs further improve global and regional forecasts with calibrated uncertainty and compact parameterization [114].

**Domain-tailored hybrids built on PIML.**    While the previous section presented evidence for extending PIML within biology, similar pressures recur outside biology as well. Relative to vanilla PIML that fits clean equations to resolved fields, many efforts introduce domain-specific adjustments that parallel BIML's pillars and can inform BIML, just as BIML can inform them. In mobility, buildings, and grids, human and operational behavior injects context and hidden drivers that clean equations do not capture; hybrid models therefore encode conservation or network laws, learn residual behavioral or operational terms, represent intent and operations as constrained latent states, and quantify uncertainty [78, 19, 127, 64, 57, 107]. In weather and climate, the difficulty is driven by multi-scale coupling, chaos, and unresolved physics; hybrid designs embed conservation and symmetry, model sub-grid processes as structured latents with uncertainty, and couple learning with physical evolution to improve extremes and large-scale forecasts [17, 8, 114]. These domain-tailored hybrids admit diffuse prior knowledge, make context explicit, elevate unobserved drivers to constrained latents, and preserve mechanistic backbones. They offer patterns BIML can adopt, and progress in BIML on uncertainty, context, latents, and scalable composition should in turn transfer back to these domains.

