# OpenReview forum: "Position: Biology is the Challenge Physics-Informed ML Needs to Evolve"
_NeurIPS.cc/2025/Position_Paper_Track — NeurIPS 2025 Position Paper Track_

### Official Review · Reviewer_mt61 · 2025-07-26

**Significance:** 3
**Presentation:** 3
**Rating:** 7
**Confidence:** 4

**Summary:**

Physics-informed Machine Learning (PIML) is an established research line, which is characterized by the integration of Physics laws into Machine Learning models.
This paper puts forward the position that a promising evolution of PIML is Biology-informed Machine Learning (BIML), which has a similar scope to PIML while presenting unique challenges.
After laying out such challenges, the position is articulated around four main pillars that should form the basis for future research in BIML, including: uncertainty quantification, contextualization, constrained latent structure inference, and scalability.
The paper discusses the possible role played by foundation models in this evolution to BIML, and it provides a short illustrative example on gene regulatory network inference.
The paper then concludes with some alternative views.

**Strengths:**

- I enjoyed reading this paper. I think it is well-written and well-structured.

- The position is clearly stated and backed up by a significant number of appropriate references.

- I believe that the topics discussed in this position paper are timely; foundation models are becoming very popular and I expect that they will play a major role in future Machine Learning approches. I found the discussion on foundation models within BIML informative.

**Weaknesses:**

I cannot find any major weaknesses in this paper.

A minor point is about Physics informed Gaussian processes and Deep Gaussian Processes, which were developed way earlier than when PIML became a thing. A few examples: Calderhead et al., NeurIPS 2008, Dondelinger et al., AISTATS 2013, Lorenzi and Filippone, ICML 2018. -- and references therein.

**Questions:**

Maybe it would have been nice to see a couple of use-cases developed to a greater extent, but I'm well aware of space limitations. Maybe something to think about for the appendix?

Would it be possible to think of other fields where PIML could evolve to similar to BIML? In other words, are there any other fields characterized by the sort of challenges specific to Biology? Maybe this could be a nice way to expand on the paragraph about "Why biology should be the next frontier for PIML". Maybe something to elaborate on in the conclusions?

**Alternative Position:**

Yes, and alternative positions are well-considered and addressed by the argument

**Author Identification:**

No.

**Context:**

3

**Details Of Ethics Concerns:**

The paper discusses Biology-Informed Machine Learning as a new field integrating Biology into Machine Learning. As far as I can tell, there are no immediate issues with Ethics concerns if this technology complies with current ethics regulations.

**Discussion:**

3

**Ethics:**

["NO or VERY MINOR ethics concerns only"]

**Position:**

Yes, the paper argues for or against a position related to machine learning.

**Support:**

3

**Thoroughness:**

4

---

### Official Review · Reviewer_rsT2 · 2025-08-04

**Significance:** 3
**Presentation:** 3
**Rating:** 5
**Confidence:** 3

**Summary:**

The paper argues that the successes of physics-informed ML (PIML) do not translate directly to biology, a domain characterised by noisy, heterogeneous, partially observed, multi-scale systems. Rather than viewing these difficulties as roadblocks, the authors frame them as an opportunity to evolve PIML into Biology-Informed ML (BIML). They identify four structural mismatches (uncertain priors, data heterogeneity, hidden variables and network complexity) and propose four corresponding pillars (uncertainty quantification, contextualization, constrained latent structure inference and scalability). Concrete recommendations include biology-centric benchmarks and incentive structures. The position is that tackling biology’s challenges will catalyse methodological advances benefitting scientific ML broadly.

**Strengths:**

- I found the presentation to be clear and accessible to both ML and computational biology audiences.
- The paper tries to make a diagnosis of why direct PIML to biology transfer fails and provides classification of four mismatches.
- Proposes actionable pillars and provides recommendations for benchmarking, incentives, interdisciplinary workshops, etc
- The paper takes a relevant position since biology is a major domain for for applying ML and the paper highlights risks of ignoring domain idiosyncrasies.
- Includes a dedicated alternative views section that tries to rebut opposing stances.

**Weaknesses:**

- The authors state "PIML ...near-complete observability" (lines 98-99) but ignores extensive work on state estimation in partially observed physical systems.
- The paper claims biology has 4 unique challenges but there might be oversimplifications. Example: climate models face uncertainty about cloud formation, aerosol interactions, and tipping points. They also involve complex networks of ocean-atmosphere-land interactions. (Social networks also exhibit similar complexity).
- The paper does not explain why biological uncertainty is fundamentally different from uncertainty in PIML. Example: cosmological simulations must handle heterogeneity across galaxies, dark matter halos and cosmic environments. Materials science also deals with heterogeneous alloys and composites. We also cannot observe dark matter directly, only its gravitational effects. The paper's claim that "only sparse glimpses" are available uniquely in biology seems to be simplistic.
- The authors claim (lines 237-238): "These...are not speculative add-ons...components." and they admit (lines 240-241) "simple baselines...outperform LLMs". This inclusion of FMs / LLMs seem to dilute the core PIML or BIML argument since LLMs have their share of issues and concerns.

**Questions:**

- How would uncertainty quantification scale to the "dozens or hundreds of interacting species" (lines 111-112)?
- Can you provide specific examples where current PIML methods have failed on biological problems due to the four challenges?
- How would you validate BIML methods when biological ground truth is often unavailable?

**Alternative Position:**

Yes, and alternative positions are well-considered and addressed by the argument

**Author Identification:**

No.

**Context:**

3

**Details Of Ethics Concerns:**

None.

**Discussion:**

3

**Ethics:**

["NO or VERY MINOR ethics concerns only"]

**Position:**

Yes, the paper argues for or against a position related to machine learning.

**Support:**

3

**Thoroughness:**

4

---

### Official Review · Reviewer_86jt · 2025-08-09

**Significance:** 3
**Presentation:** 3
**Rating:** 7
**Confidence:** 4

**Summary:**

This position paper argues that modern biology should be understood as a direct outgrowth of physics, both in conceptual foundations and methodological approaches. The authors contend that many breakthroughs in biology—particularly in molecular biology, biophysics, and systems biology—derive from physical principles and tools originally developed in physics. They advocate for a deeper integration of physics-based thinking into biological research, especially in the era of large-scale data, complex systems modeling, and AI-driven discovery. The paper reviews historical examples, outlines conceptual parallels between the disciplines, and identifies opportunities for cross-disciplinary training and research frameworks. The authors propose that embracing physics-style modeling and inference can accelerate biological discovery and improve the rigor of biological sciences.

**Strengths:**

- The central thesis is clearly articulated, with multiple historical and contemporary examples supporting the link between physics and biology.
- The paper is well-structured, progressing logically from conceptual framing to historical context and forward-looking recommendations.
- The emphasis on training, methodology transfer, and the potential for AI to strengthen cross-disciplinary research is timely.
- The writing style is clear and accessible, making it approachable for a broad NeurIPS audience.

**Weaknesses:**

- The paper’s examples, while compelling, are weighted toward molecular and systems biology; broader coverage of other biological subfields could improve generality.
- Although the argument is persuasive, quantitative evidence showing the direct impact of physics-derived approaches in recent biological breakthroughs is limited.
- The discussion of AI integration is relatively brief and could be expanded with specific scenarios or case studies.
- The call to action could be made more actionable, e.g., outlining concrete steps for the NeurIPS community to engage with biological research.

**Questions:**

1. Can the authors provide quantitative or bibliometric evidence showing trends in the adoption of physics-inspired methods in biological research?
2 How do the authors envision AI acting as a bridge between physics and biology in practical collaborative projects?
3. Are there specific subfields in biology where the physics-based approach has met resistance, and if so, how might these challenges be addressed?

**Alternative Position:**

Yes, and alternative positions are well-considered and addressed by the argument

**Author Identification:**

No.

**Context:**

3

**Discussion:**

3

**Ethics:**

["NO or VERY MINOR ethics concerns only"]

**Position:**

Yes, the paper argues for or against a position related to machine learning.

**Support:**

3

**Thoroughness:**

4

---

### Note · Authors · 2025-08-25

**1-10 Additional Comments:**

I suppose it is difficult for a brand-new track to be perfect from the start. As mentioned above, knowing beforehand that we would not have to convince reviewers would have been better. Although I would have preferred having an actual rebuttal.

Apart from that, I am happy with the process. The most important point to me is getting thoughtful and actionable reviews, I am satisfied on that part.

**1-11 Submit Again:**

Probably yes

**1-1 Submission Process:**

5

**1-3 Future Development:**

- Possibility to interact with reviewers, even shortly (not as long as for the main paper track)

**1-4 Interest:**

["Mentorship programs for early-career researchers"]

**1-5 Thoughtful:**

9

**1-6 Supportive:**

8

**1-7 Technical Aspects Versus Position:**

4

**1-8 Gate Keeping:**

10

**1-9 Camera Ready Changes:**

The bulk of the changes is about adding further details to the appendix of the paper, based on the different reviewers' suggestions. More details related to these edits can be found in the individual answers to reviewers. Broadening the scope was a common theme among reviewers, which we will happily do at camera-ready time. These additions will strengthen clarity, generality, and practical impact.

Context beyond biology.
We will add a concise related-work supplementary section showing how other domains (e.g., traffic systems, urban energy, power grids) started with Physics-Informed Machine Learning (PIML) and evolved it, thus situating our proposal, Biology-Informed Machine Learning (BIML), within a broader trend.

Where current PIML falls short in biology.
As part of the appendix, we will add concrete examples aligned with our four challenges (unsettled knowledge, heterogeneity, partial observability, scale/complexity).

Quantitative adoption trends.
We will include a brief bibliometric summary (2015–2024) using OpenAlex and a PubMed slice, to document growth in “physics-informed” terminology in biological/biomedical/health domains.

Broader biological subfields.
Extend coverage beyond molecular/systems biology (e.g., ecology, biomechanics/physiology, plant phenomics), noting where “pure physics” meets resistance and how BIML-like adaptations help.

Expanded use cases.
Beyond the Gene Regulatory Network inference example developed in Section 3.3, we'll add another case study: Generalization to unseen perturbations, e.g., patient-specific PK/PD for new drugs or temporal transcriptomic responses to unseen CRISPR KOs.

Finally, we'll edit the phrasing in the main text so that the claim that the presented challenges are unique to biology is softened, while explicitly acknowledging parallels in cosmology, climate science, and materials science. Instead, the emphasis will be put on the fact that biology *combines* these factors, stressing standard PIML.

**3-1 Review Response1:**

86jt

**3-2 Reaction To Review1:**

We thank the reviewer for a thoughtful assessment. It is aligned with our position and urges for wider scope and stronger evidence. Rather than undermining the argument, the feedback invites us to broaden context and sharpen contributions, improving generality and practical value. The tone is collegial and non-gatekeeping, with actionable suggestions on quantitative evidence and practical AI scenarios.

W1/Q2: Plant biology is one such field. A recent review [Batuwatta-Gamage et al. 2025] supports PIML but shows that dynamically changing microstructure makes the governing physics both imperfect and tightly coupled (heat/mass transport with evolving mechanics and properties), where standard PIML is sensitive. The authors call for tighter experiment–model integration and Bayesian PIML: uncertainty-aware, context-conditioned hybrids, aligned with the BIML direction.

W3/Q1.2: 1) See answer Q1@mt61 for one case study. 2) We view AI as an integrative layer letting physicists & biologists co-specify models: it distills literature/databases into readable constraints (units, stoichiometry, admissible kinetics) that physicists can encode, and turns them into hierarchical, context-aware priors (by cell type, assay, condition). AI doesn’t replace physics; it translates and adapts it to living systems so mixed teams can iterate toward models that are both mechanistically faithful and biologically specific See Sec. 3.2 and Fig. 2.

W4: Shortly, concrete steps for the NIPS community to engage with biology can be (i) developing biology-realistic benchmarks with domain experts and (ii) organizing NeurIPS workshops + challenges similar to the DREAM series.

W2/Q1.1: Using OpenAlex (Life+Health) and a PubMed-indexed biomedical slice (terms: "physics-informed"), we see a clear upward trend from 2015 to 2024: OpenAlex goes 3,23, 225 publications, while PubMed goes 28,54,341. End-to-end PIML wins in biology are rare; yet experiments show feasibility. Broader impact needs BIML.

**3-3 Review Response2:**

rsT2

**3-4 Reaction To Review2:**

Thanks for a careful, cautiously supportive assessment. It mixes position-level critique and technical queries; the tone is constructive, non-gatekeeping, and urges broader scope, evidence, and validation.

W1-2-3: We'll revise the wording to avoid implying exclusivity, e.g., by softening L98–101, explicitly acknowledging extensive PIML work on state estimation for partially observed physics or complex networked systems (climate, cosmology, materials). Thus, our claim is not uniqueness but combination: biology brings 4 challenges together that stress PIML; we’ll clarify this. Also see Q2@mt61.

W4: Thanks for pointing out the ambiguity. We’ll clarify L237–241: our point is that integration tooling for combining FMs/LLMs with PIML is emerging, while predictive gains are not guaranteed. These are complementary, not contradictory; we’ll adjust wording.

Q1: One could use Variational Bayesian Last Layers for sampling-free single-pass UQ with quadratic cost in n_output_dim for multivariate regression. Next, Gaussian score-matching VI provides closed-form moments and often 10–100x fewer gradients than Black Box VI, making high-dim Gaussian UQ practical. Alternatively, we suggest modern sampling techniques (e.g., HMC variants) exploiting sparsity/block structure of reaction networks.

Q2: Recent work on canonical dynamics (Lotka–Volterra/SIR) shows shared-ODE, parameter-shift PIML baselines degrade under functional heterogeneity; methods that let the governing form vary fare better, an explicit PIML failure mode [Discovering Physics Laws of Dynamical Systems via Invariant Function Learning, 2025]. We'll add documented cases for other challenges; space here allows only this example.

Q3: Beyond mandatory validation on synthetic data, where a ground truth is available, say for GRN inference, we will leverage downstream tasks, for which prediction error can be computed, as well as posterior predictive checks, calibration/coverage diagnosis, and prospective KO/dose checks.

**3-5 Review Response3:**

mt61

**3-6 Reaction To Review3:**

We thank the reviewer for this constructive assessment. We believe the review is supportive of the paper’s central position while offering actionable suggestions. The feedback is collegial, focused on broadening context.

W1: Thank you for surfacing this lineage. We see this GP-based constrained modeling predating "PIML" as strengthening our position, not something to replace. All 3 cited works use biological exemplars: Calderhead et al. on p53 with unobserved species; Dondelinger et al. on circadian clock regulation/signal transduction under strong noise; and Lorenzi et al. on a signal transduction cascade, with emphasis on VI-based UQ. Though small-scale (≤5 variables), they already confronted noise, unobserved species, and reliable UQ, thereby paving the way for BIML by adapting physics-derived priors to living systems. We will credit this thread in the paper & appendix.

Q1: We will add another use case - Generalizing to unseen perturbations, e.g.,
(i) patient-specific PK/PD for new drugs and (ii) temporal transcriptomic responses to unseen CRISPR KOs. Standard mechanistic/statistical models work in-distribution but falter under shift; FMs/LLMs alone add metadata and priors yet often trail simple baselines. Beyond PIML’s mechanistic/data-driven hybrid approach, BIML couples a mechanistic–probabilistic core based on the four pillars with FMs/LLMs to map new drugs/genes to related targets and pathways, supplying context-aware priors.

Q2: Thanks for this comment, which prompted us to do a literature scan beyond Biology. Several fields adopted PIML and then extended it to handle domain-specific challenges that strict physical constraints alone could not capture. Examples include traffic systems, urban energy, power grids [1-3]

[1] Physics-Informed Deep Learning For Traffic State Estimation: A Survey and the Outlook
[2] A review of physics-informed machine learning for building energy modeling
[3] Physics-Informed Machine Learning for Power Grid Frequency Modeling

---

### Meta-Review · Area_Chair_kAu9 · 2025-09-12

**Rating:** 7
**Confidence:** 3

**Strengths:**

The reviewers agreed that the paper is well-written and well-structured. It has a clear position backed up by references.
The reviewers valued the emphasis on training, methodology transfer, and the potential for AI to strengthen cross-disciplinary research.

**Weaknesses:**

Some of the challenges that the authors identify as biology challenges may not be unique to biology, according to the reviews.

One reviewer mentioned that the paper focuses more on molecular and systems biology and should cover other biological subfields to improve generality. As a response, the authors will expand their discussion to other subfields.

The discussion of AI integration is relatively brief.

**Questions:**

Can you comment a bit more about what challenges are unique to biology and what challenges are shared with other disciplines (material sciences, cosmology, etc)?

**Thoroughness:**

1

---

### Decision · Program_Chairs · 2025-09-26

Accept